# Specific Local Predictors That Reflect the Tropism of Endometriosis—A Multiple Immunohistochemistry Technique

**DOI:** 10.3390/ijms23105614

**Published:** 2022-05-17

**Authors:** Anca-Maria Istrate-Ofiţeru, Elena-Iuliana-Anamaria Berbecaru, George-Lucian Zorilă, Gabriela-Camelia Roşu, Laurențiu Mihai Dîră, Cristina Maria Comănescu, Roxana Cristina Drăguşin, Dan Ruican, Rodica Daniela Nagy, Dominic Gabriel Iliescu, Laurențiu Mogoantă, Daniel Pirici

**Affiliations:** 1Department of Histology, University of Medicine and Pharmacy of Craiova, 200349 Craiova, Romania; ancaofiteru92@yahoo.com (A.-M.I.-O.); nicola_camelia92@yahoo.com (G.-C.R.); laurentiu_mogoanta@yahoo.com (L.M.); danielpirici@yahoo.com (D.P.); 2Research Centre for Microscopic Morphology and Immunology, University of Medicine and Pharmacy of Craiova, 200349 Craiova, Romania; 3Department of Obstetrics and Gynecology, University Emergency County Hospital, 200642 Craiova, Romania; iuliaberbecaru@gmail.com (E.-I.-A.B.); laurentiu.dira@yahoo.com (L.M.D.); cristinacomanescu85@gmail.com (C.M.C.); roxy_dimieru@yahoo.com (R.C.D.); ruican.dan@hotmail.com (D.R.); rodica.nagy25@gmail.com (R.D.N.); 4Department of Obstetrics and Gynecology, University of Medicine and Pharmacy of Craiova, 200349 Craiova, Romania; 5Doctoral School, University of Medicine and Pharmacy of Craiova, 200349 Craiova, Romania; 6Department of Anatomy, University of Medicine and Pharmacy of Craiova, 200349 Craiova, Romania

**Keywords:** endometriosis, adenomyosis, immunohistochemistry, histology

## Abstract

Ectopic endometrial epithelium associates a wide spectrum of symptomatology. Their evolution can be influenced by inflammatory and vascular changes, that affect not only the structure and cell proliferation rate, but also symptoms. This prospective study involved tissue samples from surgically treated patients, stained using classical histotechniques and immunohistochemistry. We assessed ectopic endometrial glands (CK7+, CK20−), adjacent blood vessels (CD34+), estrogen/progesterone hormone receptors (ER+, PR+), inflammatory cells (CD3+, CD20+, CD68+, Tryptase+), rate of inflammatory cells (Ki67+) and oncoproteins (BCL2+, PTEN+, p53+) involved in the development of endometriosis/adenomyosis. A CK7+/CK20− expression profile was present in the ectopic epithelium and differentiated it from digestive metastases. ER+/PR+ were present in all cases analyzed. We found an increased vascularity (CD34+) in the areas with abdominal endometriosis and CD3+−:T-lymphocytes, CD20+−:B-lymphocytes, CD68+:macrophages, and Tryptase+: mastocytes were abundant, especially in cases with adenomyosis as a marker of proinflammatory microenvironment. In addition, we found a significantly higher division index-(Ki67+) in the areas with adenomyosis, and inactivation of tumor suppressor genes-p53+ in areas with neoplastic changes. The inflammatory/vascular/hormonal mechanisms trigger endometriosis progression and neoplastic changes increasing local pain. Furthermore, they may represent future therapeutic targets. Simultaneous-multiple immunohistochemical labelling represents a valuable technique for rapidly detecting cellular features that facilitate comparative analysis of the studied predictors.

## 1. Introduction

### 1.1. Clinical Characteristics and Risk Factors Related to Endometriosis

Endometriosis is one of the most common benign gynecological pathologies, characterized by the presence of heterotopic endometrial mucosa outside the uterine cavity. The locations are multiple, most commonly affecting the pelvic structures (ovaries, recto-vaginal space, urinary tract, peritoneum), along with extra-pelvic structures, such as the surgical scars, and very rarely umbilicus, lungs, or brain [1].

The exact cause of endometriosis is still unknown. Multiple mechanisms were suggested. The presence of retrograde menstruation, and coelomic metaplasia represent the basic pathogenic theories of the disease, besides vascular or lymphatic dissemination, immunological abnormalities, and iatrogenic dissemination [2]. Retrograde menstruation is a common phenomenon among women in reproductive age and is the most acknowledged theory of endometriosis. It postulates that during menstruation, endometrial fragments reach de abdominal cavity trough reversed flow and implant into the abdominal and pelvic structures, proliferate and develop ectopic endometrial implants [3]. Celomic metaplasia theory stands that the cells from the peritoneum structure are related to the Müllerian ducts and under specific circumstances may develop intro endometrial cells [4] and may represent the mechanism of endometriosis in prepubertal girls [5]. The presence of endometrial cell outside the peritoneal cavity may be explained by the lymphatic dissemination theory, that postulates that the endometrial cells travel via lymphatic vessels and veins to the lungs or diaphragm [3]. Hormonal factors and altered immunological response may stimulate the transformation of normal peritoneal cells into endometrial cells [5]. Studies showed that the leukocytes are unable to recognize the abnormal locations of the endometrial cell and potentate the disease [3]. The disease may be different women of reproductive age than in prepubertal girls due to low estrogen levels before puberty. Endometriosis may be caused by defective embryogenesis. This theory is based on findings of ectopic endometrial tissue in female fetuses and sustains those embryonic cells of the Wolffian or Mullerian ducts may persist and develop into endometriotic tissue that respond to estrogens [5]. The risk factors associated with endometriosis include early menarche, short menstrual cycle, menorrhagia, nulliparity, late menopause, and other conditions involving increased ovulatory cycles. Parity, oral contraceptives, prolonged breastfeeding, tubal ligation, and hysterectomy represent protective factors [6].

Endometriosis mainly affects women during the reproductive period, having a substantial negative impact on patients’ quality of life through the accompanying symptoms such as infertility and pelvic pain [1]. The clinical aspect is characterized by a significant variability of signs and symptoms. However, the common symptoms are dysmenorrhea, dyspareunia, chronic pelvic pain, and infertility [1]. Laparoscopic exploration and histopathological confirmation of suspect lesions represent, the gold standard in diagnosing endometriosis [1].

### 1.2. Morphopathological Features of Endometriosis

This disease is characterized by lesions that vary in appearance, size, and location. Macroscopically, ectopic endometrial foci may present as superficial “gunpowder burn” lesions on the ovaries or the visceral or parietal serosa. The lesions may be white, red, brown, or blue. They often appear as nodules or small cysts that contain old bleeding areas, surrounded by various degrees of fibrosis. Atypical or “subtle” lesions are common and may occur in the form of red implants (vesicular, polypoid, petechiae, hemorrhagic, “flame-like”) and transparent blisters [7]. In particular, we can find endometrioma on the ovary, like a cyst with a fibrous wall and a semi-fluid or dense, chocolate-like content. The malignant degeneration should be ruled out when the cyst presents wall nodules or intraluminal polypoid projections [7]. Deep, nodular infiltrative endometriosis extends >5 mm in the retroperitoneum and may affect uterosacral ligaments, vagina, bladder, ureters, or intestines. The lesions can also take the form of mucous or serous polypoid masses that imitate a neoplastic lesion [7].

Histopathological examination remains the foundation for the definitive diagnosis of endometriosis. The standard histological stains, Hematoxylin-Eosin (HE) and Masson’s trichrome (MT), highlight the endometrial tissue. It can be abnormally found either at the level of the myometrium or outside the uterine cavity. However, there are exceptions, including atypical endometriosis, in which the use of immunohistochemistry is necessary to establish an unequivocal diagnosis [8]. Microscopically, the pathology is characterized by the same tissue architecture as the uterine endometrium. The lesions consist of glandular structures surrounded by stroma, along with areas of fibrosis, old or new hemorrhages, and macrophages loaded with hemosiderin [7]. 

In addition, cellular atypia of various grades may be discovered, granting a premalignant potential for these lesions [8]. Although it is a benign pathology, endometriosis has common characteristics with malignant processes. Hence, endometrial lesions can penetrate adjacent tissues (deep infiltrative endometriosis) and be associated with neoplasms’ evolution [7]. There is a robust association between ovarian cancer and endometriosis. Common genetic mutations were discovered at the origin of both pathologies. Therefore, endometriosis-associated ovarian cancers comprise a heterogeneous group of neoplasms with a frequently favorable prognosis, including clear cell carcinoma (CCC), endometrioid carcinoma (EC), and borderline seromucous tumors (TBSM), which develop concomitantly with the endometriotic process. The diagnosis of malignant lesions established in the absence of secondary neoplasms involve identifying the structure affected by endometriosis, the neoplastic process, and the benign–malignant transition on the tissue [8].

Endometrial implants may respond to hormonal changes depending on certain factors such as the amount of stroma surrounding the glands, the degree of vascularization, and the degree of fibrosis [8].

Summarizing, the microscopic investigation of endometriosis transformations represents a hot topic in the field, with significant applicability, especially for cases with atypical clinical and imaging features and possible malignant transformation.

## 2. Objectives

In this remarkably complex histopathological context, the primary objective of this work was to identify and assess the role of the stromal changes to characterize the various locations and transformations of endometriosis. Additionally, we investigated the correlation of histological presentation, as the inflammation markers, with patients’ symptoms, especially regarding pain level.

A secondary objective of this study was to demonstrate the utility of special, multiple, simultaneous immunohistochemical staining techniques that can facilitate complete visualization of vascularization and inflammatory cells involved in the unfavorable development of the endometriosis foci, such as hyperplastic or malignant transformation, in a single field of analysis.

## 3. Results

### 3.1. Associated Symptomatology Analysis: Pain and Infertility

We noted that 51.33% of the total number of analyzed cases presented moderate and severe pelvic or abdominal wall pain. Of these cases, 14% had adenomyosis (A), 12.67% had abdominal wall endometriosis (AE), 12% patients presented ovarian endometriosis (OE) and 10% had peritoneal/pelvic endometriosis (PE). Only 2.66% of patients in the control group—for whom hysterectomy was performed for other indications and had normal endometrium experienced pain (P): 1.33% with proliferative phase endometrium (EPP) and 1.33% with normal endometrium in the secretory phase (ESP).

Of the total number of patients included in the study, 22% had spotting (S). Dividing them by groups, they were as follows: 14.67% of patients with A presented S, 2.67% of patients with PE presented S, 2% of patients with OE presented S, 1.33% of patients with ESP presented S, 0.67% of patients with AE and EPP presented S. 23.33% of patients had menorrhagia (M), of which 14.67% of patients with A presented M, 2% of each category: PE, AE and ESP presented M, and 1.33% of each category: OE and EPP presented M. 15.33% of the total patients had primary infertility (PI). Of these, 5.33% of patients with OE presented PI, 4% of patients with PE presented PI, 3.33% of patients with A presented PI, and 1.33% of each category of control groups: EPP and ESP presented PI, and patients with AE did not present PI. Additionally, 12.67% of patients associated with secondary infertility (SI) were as follows: 4.67% of patients with OE presented SI, 4% of patients of A presented SI, 3.33% of patients of PE presented SI, and 0.67% of patients with AE presented SI. The control groups did not present SI (Figure 1). Comparing the symptoms according to location, we noticed that A was associated with most cases that showed P (14% of them)/S (14.67% of them)/M (14.67% of them). In total, 12.67% of patients with AE had moderate or severe P. Additionally, 10% of patients with PE had moderate or severe P. 12% of patients with OE had moderate or severe P, but we found that 5.33% of them were associated with PI and 4.67% of them were associated with SI, representing the highest percentages associated with the categories studied.

Applying the chi-square test, we noticed that there were differences between the clinical expressions of all the symptoms analyzed as follows: the ratio between patients presenting with P/without P differs between location categories, [X^2^(1, 5 = 150) = 59.37), *p* < 0.001]; the ratio between patients with S/without S differs between the location categories, [X^2^(1, 5 = 150) = 77.73), *p* < 0.001]; the ratio between patients with M/without M differs between location categories, [X^2^(1, 5 = 150) = 70.39), *p* < 0.001]; the ratio of patients with PI/without PI differs between location categories, [X(1, 5 = 150) = 13.81), *p* < 0.001] and the ratio of patients with SI/without SI differs between location categories, [X^2^(1, 5 = 150) = 18.38), *p* < 0.005].

### 3.2. Evaluation by Classical Histopathological Staining Methods

We processed 150 surgically excised specimens, of which 100 presented endometriosis/A and 50 represented the control groups, for which hysterectomy was performed for another pathology, (Figure 1A–D) using classical histological techniques, HE, and MT. The presence of endometrial structures and adjacent stroma were diagnosed by the presence of endometrial glands and endometrial stroma, accompanied by varying degrees of fibrosis, acute or chronic hemorrhage, and lympho-monocyte inflammatory infiltration.

The ectopic epithelium presented a single layer of Műllerian type or with different architectures, but maintained the characteristics of normal endometrium. The stroma contained round/oval cells with spherical nuclei and a reduced cytoplasm. Inflammatory cells were detected around the glandular elements. The control groups were divided according to the phase of the menstrual cycle (Figure 2A,B).

In the 25 OE studied cases, we identified endometrial glands with simple columnar epithelium, periglandular stroma, and periglandular inflammatory cell infiltrate at the level of the ovarian cortex (Figure 2C).

In pelvic or peritoneal endometriosis (PE), we found ectopic endometrial glands pre-sent in the peritoneum structure, with simple cylindrical epithelium, periglandular stroma, and periglandular inflammatory infiltrate between the structures usually found in the peritoneum: mesothelium, loose connective tissue with collagen fibers, blood vessels, and fibroblasts (Figure 2D).

In A, we found endometrial glands located in the myometrium far from the normal endometrium. Nevertheless, they maintained their classic histological appearance, with round-oval glands lined with a simple cylindrical epithelium, elongated nuclei, and acidophilic cytoplasm, surrounded by rich stroma, fusiform cells, and elongated myocytes (Figure 2E). Of the 25 A cases, 12% had areas of atypical hyperplasia, and only 4% were associated with transformation into moderately differentiated G2 endometrioid carcinoma.

AE occurs after obstetrical and gynecological surgeries. Microscopically, we observed endometrial glands and stroma in the structure of the striated muscles, or in adjacent structures such as the aponeurosis or adipose tissue (Figure 2F).

Endometriosis can develop cellular atypia, as we have found in 12% of cases with A, with pleiomorphism ranging from moderate to severe, arranged in several layers, and even microscopic papillae. In addition, dysplastic lesions appear as cancer-precursor lesions with cellular atypia and architectural changes (Figure 3A,B).

Malignant transformation of endometriosis can occur in rare cases and was detected in our study in 4% of cases with A. The most common histological subtype associated with endometriosis is endometrioid carcinoma (Figure 3C).

### 3.3. Histopathological Characterization by Single Immunohistochemistry

#### 3.3.1. Cytokeratin Reactivity Analysis

First, utilizing immunohistochemistry for CK7 and CK20, we have demonstrated a CK7+/CK20- profile of the lesions, confirming that the nature of all the excised ectopic tissue was of endometrial type, and thus we excluded potential metastasis originating from the digestive tract (Figure 4A,B).

#### 3.3.2. Hormone Receptor Reactivity Analysis

The estrogen receptors (ER) and progesterone receptors (PR) were present at the level of endometrial cells in all evaluated areas (Figure 4C–F). The positive reactivity for both markers in all cases once again demonstrated the endometrial origin.

### 3.4. Stromal Vascularization Analysis

Using anti-CD34 antibody, we noticed that the highest average density was present in AE cases 37.42 (±8.13 vessels CD34 +/× 200), followed by the ESP 26.03 (±2.08 vessels CD34 +/× 200), EPP 23.54 (±2.42 vessels CD34 +/× 200), A 18.98 (±2.68 vessels CD34 +/× 200), (Figure 5A,B), PE 4.12 (±0.56 vessels CD34 +/× 200) and OE 2.69 (±0.56 vessels CD34 +/× 200)). The number of vessels showed general differences between all groups studied [F(5,149) = 477.34, *p* < 0.001]. Post hoc comparisons using the Tukey HSD test indicated a significant difference between A/AE and all of the other groups (*p* < 0.001), but the number of blood vessels labelled in cases with A was smaller than the number of blood vessels labelled in cases with EPP or ESP. The number of blood vessels labelled in cases with OE/PE were smaller compared to all other groups (*p* < 0.001). The number of blood vessels labelled in cases with EPP and ESP showed no significantly differences between them (*p* > 0.05) (Figure 2).

### 3.5. Analysis of Cell Proliferation and Involvement of Tumor Proteins

We used the anti-Ki67 antibody to study the rate of cell proliferation. We observed that the labelling for dividing cells was more pronounced in areas with A and AE than normal endometrium or endometriosis with other locations (Figure 6A, Figure 3). In the 4% of cases associated with transformation into moderately differentiated G2 endometrioid carcinoma, the mean rate of Ki67 + cells was 29.64%/× 200, and in the 12% of cases associated with atypical hyperplasia, the mean rate of Ki67 + cells ranged from 17.09%/× 200 to 13.36%/× 200 (average values were performed for each case, and then the overall average value was obtained). In cases with untransformed A, the Ki67 + cells rate ranged from 11.65%/× 200 to 4.81%/× 200. In the AE cases, the mean rate of Ki67 + cells ranged from 14.97%/× 200 to 8.18%/× 200. In PE cases, the rates of Ki67 + cells ranged from 8.01%/× 200 to 2.11%/× 200. In OE cases, the rate of Ki67 + cells ranged from 5.55%/× 200 to 3.14%/× 200, and in the EPP group, the rate of Ki67 + cells ranged from 4.11%/× 200 to 1.24%/× 200. In ESP cases, it ranged from 4.08%/× 200 to 0.93%/× 200. Ki-67 mitotic index showed global differences between all the studied groups [F(5,149) = 81.00, *p* < 0.001]. On a post hoc comparison, PE had significantly higher proliferation demonstrated by Ki67 + cells values compared to EPP and ESP (*p* < 0.05), but not OE. On the other hand, A and AE showed significantly higher proliferation demonstrated by Ki67 + cells values compared to all other groups (*p* < 0.001), but they showed no differences between them (*p* = 0.856) (Figure 3). To study neoplastic cell transformations, we used the anti-tumor protein 53 (p53) antibody to highlight the cellular expression of a tumor suppressor protein. We observed that the labelling for p53 + cells is positive only in 4% of A cases associated with transformation into moderately differentiated G2 endometrioid carcinoma the mean percentage value of p53 + cells were 58.73%/× 200, and it was necessary to make a definite diagnosis of cancer (Figure 6B).

Moreover, with the anti-B cell lymphoma 2 (BCL-2) antibody, we detected cells with potential evolution to malignancy. The expression of this altered protein had a regulatory role on programmed cell death (Figure 6C). The reaction was positive only in 12% of cases with A, associated with hyperplastic transformation and in 4% of cases with A, associated with moderately differentiated G2 endometrioid carcinoma, in which genetic changes have occurred.

Similar to anti-p53 antibody, the anti-Phosphatase and tensin homolog (PTEN) antibody were used to highlight altered cells in which a tumor suppressor gene is activated (Figure 6D). The reaction was positive only in the cases with hyperplastic transformation and G2 endometrioid carcinoma associated with A, in which more genetic changes have occurred.

### 3.6. Analysis of Perilesional Stromal Inflammatory Changes

Given that the inflammatory process may involve cellular structural and functional changes, we used several antibodies techniques to determine the presence and density of inflammatory cells around endometriosis foci with different locations compared to the normal endometrium. We used anti-Cluster of Differentiation (CD) 3 antibody for labeling T lymphocytes (Figure 7A,B; Figure 4), anti-CD20 antibody for labelling B lymphocytes (Figure 7C,D, Figure 5), the anti-CD68 antibody for labelling macrophages (Figure 8A,B; Scheme 7), and anti-tryptase antibody for mast cell labelling (Figure 8C,D; Figure 6).

We compared the results and observed that the highest mean CD3 + T-cell density was found in AE: 115.29 (±21.05 cells/× 200), followed by A: 66.02 (±6.68 cells/× 200), ESP: 15.97 (±1.54 cells/× 200), EPP: 13.43 (±1.25 cells/× 200), PE: 13.07 (±2.63 cells/× 200) and OE: 13.07 (±1.74 cells/× 200) (Figure 4).

The number of CD3 lymph cells revealed overall differences between all the studied groups (F(5,149) = 3752.43, *p* < 0.001), and post hoc comparisons using the Tukey HSD test indicated a difference between the number of CD3 + cells of the A cases and AE cases and all of the other groups (*p* < 0.001). At the same time, there were no significant differences between the number of CD3 + cells the OE cases and PE cases and between ESP cases and EPP cases (*p* > 0.05).

The average density of CD20 + type B lymphocytes varied as follows: AE: 34.42 (±5.33 cells/× 200), A: 9.59 (±1.38 cells/× 200), ESP: 4.48 (±1.6 cells/× 200), PE: 3.6 (±0.56 cells/× 200), EPP: 2.39 (±1.42 cells/× 200) and OE: 2.01 (±0.28 cells/× 200). The number of B lymph cells revealed significant differences between all the studied groups [F(5,149) = 733.85, *p* < 0.001], and post hoc comparisons using the Tukey HSD test indicated a difference between the number of CD20 + cells of the A/AE and all of the other groups (*p* < 0.001), with a gradual increase from OE to PE, to A and AE. There were also significant differences between the number of CD20 + cells of the EPP and ESP (*p* = 0.021) (Figure 5).

The average density of CD68 + macrophage cell varied as follows: A: 64.43 (±10.35 cells/× 200), AE: 56.17 (±9.87 cells/× 200), ESP: 12.18 (±1.69 cells/× 200), OE: 12.03 (±1.04 cells/× 200), PE: 11.91 (±1.06 cells/× 200) and EPP: 10.11 (±0.63 cells/× 200). The number of Macrophages revealed overall differences between all the studied groups [F(5,149) = 646.73, *p* < 0.001], and post hoc comparisons using the Tukey HSD test indicated a difference between the number of CD68 + cells of the A/AE and all of the other groups (*p* < 0.001), with a common expression level for the number of CD20 + cells of the OE, PE, EPP and ESP (*p* > 0.05) (Figure 5 and Figure 6).

The average density of mast cell Tryptase + varied as follows: A: 19.29 (± 0.88 cells/× 200), AE: 18.79 (±3.97 cells/× 200), PE: 4.69 (±1.11 cells/× 200), ESP: 3.65 (±1.08 cells/× 200), OE: 3.54 (±0.6 cells/× 200), and EPP: 2.7 (±0.46 cells/× 200). The number of mast cells revealed overall differences between all the studied groups [F(5,149) = 598.37, *p* < 0.001], and post hoc comparisons using the Tukey HSD test indicated a difference between the number of Tryptase + cells of the A/AE and all of the other groups (*p* < 0.001), with a gradual increase from OE to PE, and from EPP to ESP, but with no significant differences between these two (*p* > 0.05) (Figure 7).

After an integrated and comprehensive analysis, this study shows that the panel of immunomarkers anti-CD20, anti-Tryptase, and anti-CD34 can differentiate periepithelial stroma in endometriosis. Furthermore, each location has a unique pattern of expressing these three markers. Only lymphocytes T and macrophages do not have a particular pattern of expression depending on the location.

Further, we analyzed the correlations between the symptoms and inflammatory, vascular and proliferative features. A was accompanied in 14% of cases by P, in 14.67% of cases by S, and in 14.67% of cases by M. The symptoms were correlated with: the highest rate of Ki67 + cell proliferation (11.22%), among the types of endometrioses introduced in the study. We also found that A presented, in addition to most cases associated with symptoms and the highest percentage of perilesional inflammatory cell numbers (CD68+: 64.43± 10.35 cells/× 200, Tryptase+: 19.29 ± 0.88 cells/× 200), but also with a fairly large number of CD20 + B lymphocytes (9.59 ± 1.38 cells/× 200) and CD3 + T lymphocytes (66.02 ± 6.68 cells/× 200).

Regarding AE, we noticed that it was associated with the highest value of the number of CD34 + vessels (37.42 vessel/× 200), the highest values of the number of CD20 + B lymphocytes (34.42 cells/× 200) and value of the number of CD3 + T lymphocytes (115.29 cells/× 200) analyzing the studied groups. Additionally, AE was associated in our study with a fairly labeled number of CD68 + macrophage (56.17 cells/× 200), Tryptase + mast cells (18.79 cells/× 200) and an increased process of endometrial cells in Ki67 + cell division (10.51%). All these increased statistical values of perilesional factors were associated in our study with an increased number of patients with AE who presented P (10.46%).

OE and PE are accompanied by lower rates of immunolabelled vessels (CD34+) (OE: 2.69 ± 0.56 vessels/× 200; PE: 4.12 ± 0.56 vessels/× 200), by lower rates of proliferative cells (Ki67+) (OE: 5.55% to 3.14%; PE: 8.01% to 2.11%) and inflammatory cells (CD3+: OE: 13.07 ± 1.74 cells/× 200; PE: 13.07 ± 2.63 cells/× 200; CD20: OE: 2.01 ± 0.28 cells/× 200, PE: 11.91 ± 1.06 cells/× 200; CD68: OE: 12.03 ± 1.04 cells/× 200, PE: 11.91 ± 1.06 cells/× 200), comparative with A or AE. However, in our study, OE was associated in 5.33% of cases PI and in 4.67% of cases with SI, while PE was associated in our study with 2.67% of cases with S, in 2% of cases with M, and in 4% with PI, lower values compared to groups A and AE.

Regarding the control groups EPP and ESP, we noticed that they were associated with lower values of the number of inflammatory cells (CD68+: EPP: 10.11 ± 0.63 cells/× 200, ESP: 12.18 ± 1.69 cells/× 200; Tryptase+: EPP: 2.7 ± 0.46 cells/× 200, ESP: 3.56 ± 1.08 cells/× 200, compared with A or AE groups; rates of proliferative cells was lower (Ki67+: EPP: 4.11% to 1.24%, ESP: 4.08% to 0.93%), and rates of immunolabelled vessels were lower (CD34+: EPP: 23.54 ± 2.42 vessels/× 200, ESP: 26.03 ± 2.08 vessels/× 200), comparative with A or AE groups. These groups (EPP and ESP) did not present significant differences between them and were rarely associated with P (EPP: 1.33%, ESP: 1.33%), S (EPP: 0.67%, ESP: 1.33%), M (EPP: 1.33%, ESP: 2%), PI (EPP: 1.33%, ESP: 1.33%) and SI (EPP, ESP: 0%) (Figure 1, Figure 2, Figure 3, Figure 4, Figure 5, Figure 6 and Figure 7).

### 3.7. Histopathological Characterization by a Novel, Multiple Immunohistochemical Labeling Technique

Lastly, we developed a multiple simultaneous immunohistochemistry standardized protocol of simultaneous immunohistochemistry through which we performed an immediate and precise investigation of the epithelial changes of endometriosis. Through this protocol, we demonstrated that CK7 may be useful in diagnosing endometrial epithelium and evaluating ectopically epithelium. Simultaneous visualization of macrophages (CD68 +), mast cells (Tryptase +) and T lymphocytes (CD3 +) provided a complete field of view for inflammatory elements. Therefore, as shown above, the essential stromal characteristics that characterize each type of endometriosis can also be visualized and evaluated in an integrated manner, with minimal loss of pathological material (Figure 9A–I).

## 4. Discussion

The need to investigate and research tissue changes associated with endometriosis and adenomyosis has increased, due to the rising incidence and improvement of diagnosis techniques during the last decades [8,9,10,11].

Premalignant transformation, complex atypical hyperplasia, and malignant transformation of endometriosis foci can be highlighted by identifying the typical glandular structure in malignant areas by classical histological staining and later confirmed by immunolabeling to accomplish the differential diagnosis [8].

Rolly [12] published the first case of malignantly transformed adenomyosis in 1897. Several hyperplastic or neoplastic transformed adenomyosis cases have been reported [12,13], but this pathology remains rare in women of reproductive age. In our study, we found atypical endometrial hyperplasia in 12% cases of A and malignant transformation in 4%. Thus, we can support Rolly’s [12] assertion that we seldom encounter these changes.

Even if adenomyosis is a benign pathology, it has some aspects of malignancy: it proliferates, angiogenesis is quick and abundant, and it has an invasive potential. The transition of the endometrial epithelium from the structure of the adenomyosis to the premalignant tumor structure and later to the carcinomatous lesion of different degrees were described. These transformations depend on the assembly of genetic and epigenetic changes as a continuous process. However, the identity of the molecular substrate that initiates the transformation of myometrial cells and regulates their growth remains unknown [13].

### 4.1. Evolution Predictors for Endometriosis/Adenomyosis Foci

#### The Role of Cytokeratin

In this study, we demonstrate the presence of endometrial epithelium and ectopic endometrial glands in the structure of various organs. We immunolabeled the epithelium with specific endometrial cytokeratin (CK7) [14,15,16], the periglandular vascularization involved in tumor growth, and the tumor proteins involved in the process of hyperplastic transformation. The endometrial epithelial cells typically express this cytokeratin. CK7 is also used for differential diagnosis with a tumoral process of digestive origin, where the immunolabeling is negative for CK7 and positive for CK20 [14]. However, it may also be positive in gynecologic malignancies with an ovarian origin, and the evolution of these pathologies is aggressive, compared to endometriosis/A [17].

### 4.2. The Role of Hormone Receptors

We observed that the ERα and PR responses were positive in all the studied ectopic endometrial foci, which proves once again that the studied areas were foci of endometriosis/adenomyosis, which were controlled and influenced by hormonal mechanisms. ERα is a nuclear receptor activated by the sex hormone estrogen and encoded by the Estrogen Receptor 1 gene (ESR1) [18,19]. It determines the activation of cell proliferation in the uterus [20].

The progesterone secreted in the body activates PR, changes DNA structure, and influences cellular multiplication. The presence of steroid receptors in endometriosis or A tissue suggests that they are hormone-dependent to varying degrees. This assumption is supported by the presence of symptoms and cyclical signs of endometriosis [21]. However, in our study, although all histopathological areas analyzed were positive for PR, not all patients developed symptoms.

### 4.3. The Role of Perilesional Vascularization

CD34 was initially described in hematopoietic stem cells [22,23,24] and acts as an adhesion molecule that contributes to the movement of T lymphocytes to the lymph nodes, binding specifically to the T cell [25,26]. Moreover, it can block the action of eosinophils and mast cells locally [27,28]. Using immunohistochemical labeling, we identified localized periglandular neoformation vessels with a role in tissue growth and development, thus contributing to endometrial cell proliferation and highlighting the possible occurrence of endometriosis by vascular dissemination [11]. We noted that in the AE case, the perilesional vascular density was the highest, followed by ESP, EPP and A. This may have significant clinical implications such as the accelerated increase of the endometriosis areas.

### 4.4. The Role of Cell Proliferation and the Presence of Tumor Proteins

In humans, Ki67 represents a cell proliferation marker [29,30]. It was intensely positive in cases with uncontrolled cell division, progressing to hyperplastic or malignant transformation of the studied areas [30,31]. In our study, we observed that the density rate of Ki67 + positive cells was highest in neoplastic transformed A associated with (moderately differentiated G2 endometrioid carcinoma), followed by cases of hyperplastic transformed A and then cases of AE, PE, OE, EPP and ESP, thus demonstrating that the most susceptible cases for transformation in our study are those with A and AE.

The tumor suppressor protein p53, described in the literature as “genome guardian”, is involved in apoptosis, inhibiting angiogenesis, and maintaining genome stability. p53 can repair damaged Deoxyribonucleic acid (DNA) [32] and is involved in cell aging by blocking cells at different cell cycle stages to create enough time to repair DNA. It can also initiate apoptosis when irreparable DNA damage occurs [32]. In our study, we observed that the mean rate of p53+ positive cells was highest in A associated with moderately differentiated G2 endometrioid carcinoma, followed by cases of hyperplastic transformed A and all the other groups were negative. These positive areas of A associated with hyperplasia have escaped genomic control and may become malignant in the future.

BCL-2 is an oncoprotein that acts to regulate cell death (apoptosis) by inducing it (pro-apoptotic effect) or inhibiting it (antiapoptotic effect) [33]. In the endometriosis/adenomyosis cases, immunohistochemical labeling techniques showed that ectopic endometrial cellularity reacts positively, supporting cell death inhibition with hyperplastic or neoplastic transformation. Moreover, according to several studies, these epithelial protein changes cause fast and chaotic evolution with transformative potential [34]. In our study, the A cases associated with G2 endometrioid carcinoma and hyperplastic transformations were positive for BCL-2. The rest of the cases were negative.

PTEN is a tumor suppressor protein that has pushed research into several cancers’ development by inhibiting it. It usually performs its function through a protein phosphatase, which regulates the cell cycle, preventing accelerated growth and multiplication [35]. Mutations in the gene encoding this protein block the enzymatic action, further increasing the rate of cell proliferation and decreasing the rate of cell apoptosis. We observed a positive immunohistochemical reaction to the anti-PTEN antibody in endometriosis samples, only in the hyperplastic transformed and the G2 endometrioid carcinoma areas, while the rest of the cases were negative.

Deep infiltrative endometriosis can mimic a malignant lesion by the polypoid or mucous appearance that penetrates adjacent tissues [11]. The immunolabeling with anti-p53, anti-BCL2 and anti-PTEN antibodies can differentiate this benign lesion from an infiltrative, malignant lesion.

### 4.5. The Implications of Perilesional Inflammation

Inflammatory factors secreted by the periglandular immune system cells, can influence the evolution and transformation of the abnormal cells, with carcinogenetic implications. Thus, inflammation can lead to atypical aspects of endometrial cells, altering the nucleocytoplasmic ratio, the nuclei can become hyper- or hypochromic, and the cell layers can multiply [8]. The inflammatory cells invading the stroma and can influence the transformation of this benign pathology into a hyperplastic or even malignant pathology. This inflammatory microenvironment associated with endothelial dysfunction participates in carcinogenesis [8].

The intense perilesional inflammatory response plays a double role, in defense and creating an environment favorable to hyperplastic transformations [8].

CD3 is expressed in the cytoplasm of pro-thymocytes and thymus stem cells that further develop into identified mature T cells, histologically identified using anti-CD3 antibody immunostaining techniques. The exaggerated numerical increase of T lymphocytes supports a significant inflammatory process or pathology with unfavorable evolution, hyperplastic, or even malignant [36]. Around the foci of endometriosis and A, we observed a considerably increased number of CD3 + T lymphocytes in areas with A with or without hyperplastic/malignant changes and thus demonstrating their involvement in tumor proliferation [20].

CD20 was expressed on the surface of B lymphocyte cells and facilitates the interaction of the B cell microenvironment, further involved in developing various malignancies [37]. However, around the A (being associated with 12% cases of hyperplasia and 4% cases of malignant transformation in our study) and AE, their number was higher, compared to the other groups included in the study.

CD68 was an intensely expressed protein in monocytes, circulating macrophages, or tissue macrophages (microglia, Kupffer cells). This was a transmembrane glycoprotein identified in foci of peritoneal endometriosis in many previous studies [37,38]. It has a role in mediating and exacerbating inflammation and supporting the microenvironment of this pathology [39,40]. The macrophages increase the secretion and synthesis of local proinflammatory mediators, and this can define the stability and maintenance of endometriotic foci with their accelerated growth and proliferation [38,39,40]. Macrophages play an essential role in the destruction, repair, and regeneration of endometrial tissue [41], especially in the menstrual and proliferative phases of the endometrial cycle.

Similar to malignant tumors, A shows a growth that produces local invasion, and in vitro studies have shown specific mechanisms that have influenced tissue transformation [42]. We observed sporadic mast cells or even absent around the proliferating and secretory endometrium. However, their number increases within the lesion, which shows that the local functional balance has been disturbed. Mast cells produce a wide range of proteases and enzymes, one of which is Tryptase, used as an immunohistochemical marker in cell identification [43]. We found a higher density of these cells in endometriosis foci than in the normal endometrium. This aspect plays an essential role in the fibrogenesis processes. It initiates a local inflammatory response by releasing several mediators such as proteases, histamine, cytokines: IL-1, IL-6, IL-8, TNFα, TGFβ, macrophage granulocyte colony-stimulating factor (GM-CSF). In addition, the cytokines IL-1α, IL-6, IL-8, IL-18, and TNFα have been found to increase in endometriosis [40,44,45] consistently. All these molecular elements maintain a solid inflammatory process around the foci of the ectopic endometrium, which predisposes to epithelial transformation.

Regarding perilesional microvascularization, we observed that the highest *vessels density* was identified in AE and A cases. The results obtained were similar to those of Schindl M. et al., who showed that in the case of A, the number of CD34 + vessels are higher compared to the control endometrium [46].

We observed that the number of *CD68* + macrophages is lower in ESP and EPP than A/AE or than the rest of the groups including in this study contrary to the findings of Khan et al., that there was an increase in macrophage populations in both ESP and EPP of the endometrial cycle [47].

We also found that the density of mast cells is higher in the A and AE cases com-pared to the rest of the categories. Similar findings were reported [48,49], that there is an increased perilesional number of mast cells and no variations in the endometrium of patients with endometriosis.

### 4.6. The Role of Stromal Factors in Symptomatology: Pain and Infertility

Following the analysis of inflammatory, vascular, hormonal, proliferative and tumor factors, we observed that A has the most intense stromal changes (number of macrophages, mast cells). It is associated with the highest degree of cell proliferation, with increased invasive potential, with the most common changes in tumor proteins (p53, BLC-2, PTEN) and is most often accompanied by signs and symptoms (P, S, M) in our study. The intense proliferative and inflammatory modifications were associated with violent symptomatology, which suggest the correlation between the degree of cell proliferation, the appearance of local inflammation and the onset and severity of symptoms [11,31,46,48]. Additionally, in our study A was associated with a higher degree of cell proliferation. An explanation for this association is that cell proliferation is supported by local inflammatory elements, which also increase cell adhesion, and may predispose to pelvic growth and adhesion, associated with deep infiltrative endometriosis (DIE) and increased severity of the symptoms (P, S, M) [49]. In our study, OE is less associated with inflammation and symptoms, but is strongly correlated with primary and secondary infertility. In the cases of AE included in our study, compared to other groups, we can observe that the most accentuated local vascularization was associated. AE was also associated with significant perilesional inflammation (B lymphocytes and T lymphocytes) and a significant degree of cell proliferation, compared to other groups, but with lower values than A. These local aspects lead to the growth tumor, with consecutive local mechanical compression, accentuation of the inflammatory processes and the consecutive cascade of local reactions. These reactions are triggered by the CD3 +, CD20 +, CD68+ and Tryptase+ high cellularity, which cause significant local pain [11,13,46,48,50]. PE was accompanied by a low perilesional inflammation, cell proliferation and signs and symptoms (S, M), compared to other groups; however, they were associated with the highest percentage of PI and SI, compared to the other groups included in our study. Similar studies have shown that endometriosis can be associated with infertility through several mechanisms such as changes in the structure of the pelvic anatomy, hormonal and ovulatory disorders, impaired peritoneal function by inflammation, impaired hormone and endometrial cell function [51]. Adherence syndrome following pelvic inflammation can disrupt tubal permeability, preventing the release and transport of oocytes [52]. The inflammatory changes of the periglandular stroma stimulate tumor development and support the invasiveness process. Furthermore, moderate and severe pain—Numerical Rating Scales (NRS) > 6 [53], is determined by the accentuated inflammatory process and the mass increase of the ectopic tissue with accentuated tissue proliferation rate, which creates a local compression effect [11]. S and M can occur through the involvement of estrogenic and progesterone factors that promote glandular growth and development [11].

### 4.7. The Role of the Novel Technique of Multiple Immunolabeling

Multiple simultaneous immunolabeling technique helps identify and differentiate cellular and vascular elements on the same tissue slide; as a result, we can obtain a comparative study of the same slide and save significant resources. As observed in previous studies, simultaneous immunolabeling may provide concrete comparative studies of vascularization and immunolabeled inflammatory cells in the same field of interest, without varying by section [51,54]. Through the special immunolabeling method, the essential stromal features that characterize each type of endometriosis can also be visualized and evaluated in an integrated way, with minimal loss of pathological material. Simultaneous visualization of macrophages (CD68 +), mast cells (Tryptase +) and T lymphocytes (CD3 +) allowed simultaneous quantification of stromal predictors, all in a single slide.

## 5. Materials and Methods

All the patients with endometriosis involved in this study (Craiova, Romania) were surgically treated in our clinic. Informed consent was obtained from all subjects. The study was conducted according to the guidelines of the Declaration of Helsinki and approved by the Committee of Ethics and Academic and Scientific Deontology of the University of Medicine and Pharmacy of Craiova (UMFCV)(no. 88/13 September 2018). The tissue acquired during the surgery was bathed in 10% formalin solution for fixation and embedded in paraffin.

We analyzed 100 cases: 25 patients with ovarian endometriosis, 25 patients with pelvic endometriosis, 25 patients with adenomyosis (12% had areas of atypical hyperplasia, and only 4% were associated with transformation into moderately differentiated G2 endometrioid carcinoma), and 25 patients with abdominal wall endometriosis. In addition, we studied normal endometrial tissue samples from 50 hysterectomy specimens including 25 samples with proliferative endometrium and 25 with secretory endometrium (control group). All these hysterectomies were performed for uterine polyfibromatosis with surgical indication.

The patients signed a consent form for personal data use. All tissues samples analyzed were acquired in the 2nd Clinic of Obstetrics-Gynecology, Clinical County Hospital of Craiova.

### 5.1. Assessment of Patient Symptoms and Infertility

All patients completed a form stating whether their symptomatology associated pelvic or abdominal wall pain, (NRS) > 6 (moderate and severe), spotting, menorrhagia, and/or primary and secondary infertility [54].

### 5.2. Histopathological Preparation

The histology research facilities were provided by the Research Center for Microscopic Morphology and Immunology, University of Medicine and Pharmacy of Craiova.

The tissue samples were acquired and diagnosed between 2018–2021. We performed the histological preparation and examination at the Histology Department of the UMFCV. After fixation of the biological material in 4% neutral buffered formalin, the tissue fragments were routinely processed for paraffin embedding and sectioning as 4 µm-thick sections [55].

The diagnosis was based on histopathology and immunohistochemistry techniques. Histological staining for HE and MT was utilized to ascertain the diagnosis of endometriosis.

The immunohistochemistry technique has a standard algorithm, with variations depending on the antibodies used. First, the slides underwent antigen retrieval by microwaving in a specific pH 6 0.1 M citrate buffer or pH 9 Ethylenediaminetetraacetic Acid (EDTA) Solution, as specified by the producers (Table 1). The slides were incubated in a 3% Hydrogen Peroxide Solution of 3% for 30 min to block endogenous peroxidase that could interfere with signal detection, and then in 3% skimmed milk saline solution to block the antibody nonspecific binding sites. Next, the primary antibodies were incubated on the slides for 18 h in the refrigerator at 4 °C, diluted as optimized (Table 1; Dako, Glostrup Denmark and Abcam, Cambridge, UK). The next day, after thorough washing, the slides were incubated with HRP-labelled secondary antibodies specific for the species of the primary antibodies (Nikirei-Bioscience, Tokyo, Japan). The last step of the reaction was the actual detection of the signal utilizing 3-3′ diaminobenzidine (DAB, Nikirei-Bioscience), after which the sections have been counterstained with Mayer-hematoxylin, dehydrated, clarified in xylene, and covered with Canadian Balm for imaging and analysis.

### 5.3. Novel Technique for Multiple Immunohistochemical Labeling

We developed an immunohistochemistry complex technique to evaluate endometriosis and A with and without hyperplastic/malignant transformation. The protocol is based on previously reported techniques by Pirici, D. et al. [52] and Istrate-Ofițeru, A.M. et al. [47], with some improvements. Thus, the classical protocol, as described above, was first followed for detecting the first primary antibody (mouse anti-CK7) with a goat anti-mouse secondary antibody for 2 h (Vector ImmPRESSTM HRP, MP-7452, Vector Laboratories, Burlingame, CA, United States), and signal detected with the Histofine DAB-3Skit (Nichirei-Histofine). The following day, after visualizing the glandular tissue stained in brown, the sections were incubated for 1 h in an acidic solution pH = 2 (50 mL Sodium Dodecyl Sulfate (SDS, Sigma-Aldrich, St. Louis, MO, United States) 20% + 1.876 g glycine + 1l dH_2_O) (Sigma-Aldrich) preheated to 60 °C, to elute the previously applied antibodies. The slides were then washed in 1% Phosphate-Buffered Saline (PBS, Sigma-Aldrich) solution, and the second primary antibody (mouse anti-Tryptase) was added to the slides, incubated overnight, and detected the next day with the goat anti-mouse secondary antibody for 2 h (Vector ImmPRESSTM HRP, MP-7452).

Tryptase was next visualized using the ImmPACTTM VIP Peroxidase Substrate Kit (SK-4606, Vector Laboratories), mast cells being immunolabeled in purple. The slides were then washed in PBS solution and placed again in the SDS acid solution. After thorough washing in PBS, we added a mixture of appropriately diluted mouse anti-CD68 + rabbit anti-CD3 primary antibodies (Table 1). On the fourth day, we amplified their signals with an anti-mouse Alkaline-Phosphatase Polymer (Nikirei-Histofine) for one hour and an anti-rabbit Peroxidase Polymer (Nikirei-Histofine) for another hour. The anti-CD68 antibody was developed with a Fast Red Alkaline phosphatase chromogen (Nikirei-Histofine, code: 415161F) for 20, which colored the macrophages in red-pink, and the anti-CD3 antibody was visualized using the Permanent HRP Green Kit (Zytomed, ZUC070-100, Zytomed Systems GmbH, Berlin, Germany) for 20 min and the lymphocytes T stained in green. Subsequently, the nuclei were counterstained with Mayer Hematoxylin, slides rewashed in distilled water, and put in an aqueous medium (Vecta Mount^TM^ AQ—Aqueous Mounting Medium H-5501, Vector Laboratories).

Images were captured utilizing a Nikon Eclipse 44i microscope equipped with a Nikon DS-5Mc (Nikon Europe BV, Amsterdam, The Netherlands) 5 megapixel cooled charged-coupled device (CCD) and controlled by the Image ProPlus AMS 7 (Media Cybernetics, Rockville, MD, USA) image analysis package. We analyzed the images obtained and observed the involvement of the inflammatory process in the unfavorable evolution of the foci of hyperplastic transformed adenomyosis.

### 5.4. Analysis Methods

Symptoms analysis. Patient answers were quantified as a percentage in Microsoft Excel and analyzed by location to analyze of symptoms according to the location of endometriosis. For each symptom, depending on the location of the endometriosis, the chi-square test was applied using the same program, and the values obtained were explained in the text.

Histological measurements. An average of 4 images have been captured in the lesional areas and periglandular (at the same distance from the endometrial glands), with the 20× objective from each specimen utilizing constant manual exposure and illumination settings. All elements (cells, blood vessels, cells in division) have been counted manually utilizing the “manual tag” feature in Image ProPlus, averaged on each slide, and then for each pathological subgroup. In order to quantify the number of Ki67 + nuclei (nuclei in division) and the number of p53 + nuclei, we also analyzed 4 fields/case. We counted the total number of nuclei in the glandular epithelium and the number of positive nuclei and expressed them into percentages (%) using Microsoft Excel. The final percentage averages were plotted. For the analysis of PTEN and BCL-2, cases were noted as positive and negative and were represented as a percentage of the total number of cases with similar localization. Data have been plotted in Microsoft Excel sheets, and statistical analysis was further performed using SPSS and univariate analysis of variance (ANOVA) with the Tukey’s HSD (honestly significant difference) post hoc analysis with multiple comparisons (significance set at *p* < 0.05), for continuous data. Chi Square test had been utilized for categorical data. The figures display the mean value and standard deviations (SD).

## 6. Conclusions

The extensive study of endometriosis perilesional configuration is essential as it helps establish the disease prognosis.

Under the influence of certain mechanical, hormonal/vascular factors and local inflammatory processes, the endometrial cells from different endometriotic foci can undergo different transformations. All these local factors involved in increasing the invasiveness of endometriosis, associated with exacerbation of patient symptoms, may represent therapeutic targets in the future.

The increased vascularization induces a significant tumor proliferation, with local mechanical changes that increase the perilesional inflammatory process and significant changes in the histoarchitecture of the microenvironment adjacent to the foci of endometriosis or A. Thus, cellular transformations occur especially in cases of A or AE. Therefore, rapid determination of changes in the foci of endometriosis can dictate therapeutic approach. The areas associated with atypical hyperplasia/malignancy of endometriosis can be highlighted by specific immunolabeling techniques that reveal the inactivation of tumor suppressor genes and the blocking of programmed cell death.

Simultaneous multiple immunohistochemical labelling represents a valuable technique for rapidly detecting cellular features that improve histopathological diagnosis and facilitate comparative analysis of the studied predictors.

## Data Availability

All data presented here are available from the authors, upon reasonable request.

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
