# Peer review of "Specific Local Predictors That Reflect the Tropism of Endometriosis—A Multiple Immunohistochemistry Technique"

_ijms, 2022, doi:10.3390/ijms23105614_

Round 1
Reviewer 1 Report
The study shows interesting findings including various types of endometriosis, and adenomyosis furthermore. The authors included various type of staining, antibodies.
However, Result and Discussion section need modifications, overall. The presentation is scattered, not organized well. It is rather confusing, not showing its strength enough.
Additional comments:
The manuscript-especially the result /discussion section needs to be re-written. They aren't organized well, scattered in too many paragraphs.
The result section seems not revealing their results of experiments enough. I suggest to organize the section thoroughly, in neat manner.
Author Response
Author's Reply to the Review Report (Reviewer 1)

Reviewer 2 Report
This paper is exciting, but the difficulty is that it is difficult to know whether the overall point is to diagnose ectopic endometriosis or malignant transformation. For example, the high expression of p53, PTEN, and BCL2 is unreliable, especially since there is no simultaneous pathological or clinical information indicating that the PTEN-high specimen is Hyperplastic transformed adenomyosis. Much information and long sentences have been written, but I have no idea what the point of this paper is.
Author Response
Author's Reply to the Review Report (Reviewer 2)

Round 2
Reviewer 1 Report
The study includes interesting results using multiple, simultaneous immnunohistochemistry method to reveal meaningful markers to identify endometriosis with unfavorable prognosis-such as atypical endometriosis, malignancy.
- The scatterness of the manuscript has been improved comparing to the previous version. However, each paragraph is composed with incompatible sentences still. Importantly, although the manuscript was modified thoroughly, I feel it is uneasy to understand the logic of the study-how aim/method/conclusion meets.
- In introduction, please provide supporting evidence of the definition of eutopic endometrium (page 2 line 88-89). I cannot agree with the definition, personally.
- In Result, multiple results from the exams are listed. However, they are not directing/organized to lead a conclusion. Many, various IH staining exams were performed: cytokeratin reactivity, perilesional vascularization, ki67 mitotic index, perilesional inflammatory changes, CD 20 density, CD68, mast cell density. In most of them, adenomyosis or abdominal wall endometriosis revealed high percentage (statistics not indicated-unknown). Is it possible to say that A or AE has an increased risk for atypical endometriosis or cancer? Moreover, any report of atypical endometriosis among the study subjects? Any correlation between atypia/or malignancy and location of EMS, or staining method?
- In page 4, correlation with the location of endometriosis and various symptom has been presented. a.Any statistical significance among the differences/comparsions? b. Comparing the pain, pelvic adhesion and presence of DIE will affect, additionally. Can it be justified to compare the location of endometriosis plainly?
Round 3
Reviewer 1 Report
The manuscript has been improved very much.